

# Carbon mapping in pine-oak stands under timber management in southern Mexico

Ashmir Ambrosio-Lazo[1], Gerardo Rodríguez-Ortiz[1], Joaquín Alberto Rincón-Ramírez[2], Vicente Arturo Velasco-Velasco[1], José Raymundo Enríquez-del Valle[1] and Judith Ruiz-Luna[1]

[1] Division of Postgraduate Studies and Research, National Technological Institute of Mexico/Technological Institute of the Valley of Oaxaca, Ex Hacienda de Nazareno, Santa Cruz Xoxocotlán, Oaxaca, Mexico
[2] Campus Tabasco, Ciencia Ambiental, Postgraduate College, Cardenas, Tabasco, Mexico

## ABSTRACT

The destructive and empirical methods commonly used to estimate carbon pools in forests managed timber are time-consuming, expensive and unfeasible at a large scale; satellite images allow evaluations at different scales, reducing time and costs. The objective of this study was to evaluate the tree biomass (TB) and carbon content (CC) through satellite images derived from Sentinel 2 in underutilized stands in southern Mexico. In 2022, 12 circular sites of 400 m$^2$ with four silvicultural treatments (STs) were established in a targeted manner: 1st thinning (T1), free thinning (FT), regeneration cut (RC) and unmanaged area (UA). A tree inventory was carried out, and samples were obtained to determine their TB based on specific gravity and CC through the Walkey and Black method. The satellite image of the study area was downloaded from Sentinel 2 to fit a simple linear model as a function of the Normalized Difference Vegetation Index (10 m pixel$^{-1}$) showing significance ($p \leq 0.01$) and a adjusted $R^2 = 0.92$. Subsequently, the TB and CC (t ha$^{-1}$) were estimated for each ST and managed area. The total managed area (3,201 ha$^{-1}$) had 126 t TB ha$^{-1}$ and 57 t C ha$^{-1}$. Of the areas with STs, the area with FT showed the highest accumulation of TB (140 t ha$^{-1}$) and C (63 t ha$^{-1}$) without showing differences ($p > 0.05$) with respect to those of the UA, which presented 129 t TB ha$^{-1}$ and 58 t C ha$^{-1}$. The satellite images from Sentinel 2 provide reliable estimates of the amounts of TB and CC in the managed stands. Therefore, it can be concluded that an adequate application of STs maintains a balance in the accumulation of tree C.

Corresponding author
Joaquín Alberto Rincón-Ramírez, jrincon@colpos.mx, jrinconr@gmail.com

## INTRODUCTION

In Mexico and globally, the carbon sequestration (C) provided by forests has become increasing relevant due to the high emissions of $CO_2$, which cause a climate imbalance and negative effects on society (*Yu et al., 2022*). At present, the destructive and empirical methods commonly used to estimate forest C are time-consuming, expensive and not feasible at a large scale (*Amiri & Pourghasemi, 2022*; *d'Oliveira et al., 2020*; *Muhe & Argaw,*

*2022*). For this reason, the use of satellite images obtained through remote sensing helps evaluate C reserves at different spatial and temporal scales, providing accurate data for strategic forest management plans (*Dou & Yang, 2018*; *Vashum & Jayakumar, 2012*) and serving as a key factor for the successful implementation of C market mechanisms (*Herold et al., 2011*).

Remote sensors allow to get information of the vegetation state, through vegetation indices (VI) that are generated with the combination of spectral bands; these VI are correlated with several variables, such as biomass, density, volume and C (*Isbaex & Coelho, 2021*). These variables are estimated by adjusting allometric models with data collected in the field based on the vegetation indices (*Chen et al., 2018*; *Pandit, Tsuyuki & Dube, 2018*; *Pertille et al., 2019*). These multiscale estimation models are commonly used in forest vegetation studies, although they are also applied in various nutrient monitoring studies to other types of vegetation (*Chen et al., 2018*; *Dou & Yang, 2018*).

The European Space Agency launched the Sentinel 2 platform to provide services based on observations of high-resolution multispectral information of the Earth surface (*Drusch et al., 2012*). With this information, it is possible to monitor changes in forests and land cover and therefore manage natural disasters (*Wong, Fung & Yeung, 2019*). Sentinel 2 is an optical satellite with 13 spectral bands that was launched in 2014. The spatial resolution of the bands varies between 10 and 60 m. In addition, it has a coverage area of 290 km and revisits the same area over a short time period (*ESA, 2021*). The freely available high-resolution Sentinel 2 satellite datasets have created new possibilities for mapping and monitoring different ecosystems and vegetation types (*Hudait & Patel, 2022*; *Puletti, Chianucci & Castaldi, 2018*).

The normalized difference vegetation index (NDVI) is the most widely used method to increase the differentiation in vegetation in remote sensing data (*Chu et al., 2019*). It is obtained by calculating the red and near infrared bands together and increases precision when used in the classification of green areas (*Pettorelli, 2013*). The NDVI is one of the indicators commonly used to detect and indicate the status and dynamics of vegetation cover (*Wong, Fung & Yeung, 2019*; *Xing et al., 2020*). Some vegetation index (VI) to evaluate biomass and C, similar to the NDVI and that use the red and NIR bands are: the Soil Adjusted Vegetation Index (SAVI), which uses an adjustment factor to the bottom of the canopy, explaining the difference between red and near infrared, this is mostly used in areas with low vegetation (*Karnieli et al., 2001*; *Sonnenschein et al., 2011*); and the Advanced Vegetation Index (AVI), which uses an adjustment factor to observe the variations of the forest over time (*USGS, 2019*). There are others VI to evaluate vegetation cover, among them is the Green Normalized Difference Vegetation Index (GNDVI), which uses the green band and NIR, it is used to make estimates of photosynthetic activity, but it is mostly applied to estimations of water and nitrogen consumption of the vegetal cover (*Ihuoma & Madramootoo, 2020*); and the Enhanced Vegetation Index (EVI), which helps to estimate biomass and C, it uses the NIR, red and blue bands, correcting some atmospheric saturations, being sensitive in areas with dense vegetation (*Choubin et al., 2019*; *Jensen, 2015*).

Aboveground biomass plays an important role in the C cycle at local and regional levels, which is why quantifying and monitoring it through satellite images are important (*Main-Knorn et al., 2013*; *Puliti et al., 2021*). Quantifying the TB and C contents (CC) of forests under timber management is important for understanding their dynamics, evaluating the effect of applied silvicultural treatments (STs) and making decisions about the sale of C credits by the community (*Cutini, Chianucci & Manetti, 2013*; *d'Oliveira et al., 2020*; *Joshi & Dhyani, 2019*; *Rodríguez-Ortiz et al., 2019*).

San Juan Lachao started its first timber management program in 2010, with an authorized area of 2,359.6 ha. In 2022, it expanded to 3,201 ha that have been part of the voluntary carbon credit market. Thus, it is important to estimate the effect of timber management and its STs in relation to C capture through satellite images since this approach provides viable information at a large scale and is an important data validation tool. In addition, in southern Mexico, there are no studies of this type. The objective was to evaluate the TB and CC of San Juan Lachao, Oaxaca, Mexico, through vegetation indices derived from satellite images. Given the hypothesis, there are no significant differences ($p > 0.05$) between the amounts of C estimated with spectral data and those estimated in the field.

## MATERIALS & METHODS

### Study area

The research was carried out in a forest under the management of San Juan Lachao, Pueblo Nuevo, Juquila, Oaxaca, Mexico, with coordinates of 16°09′30.26″N and 97°07′28.04″W and an average altitude of 1,900 m (Fig. 1). The predominant climate is warm subhumid (Cw), with an average annual temperature of 22 °C and an average rainfall of 2500 mm. The community of San Juan Lachao has a pine-oak forest of 3,201 ha, of which 573 ha are under forest management through the silvicultural development method (SDM) and 1,787 ha are under the Mexican Management Method for Uneven-aged Forests (MMOBI). This area contains species of timber importance, such as *Pinus douglasiana* Mtz., *P. maximinoi* HE Moore, *P. devoniana* Lindl., *Quercus rugosa* Née, *Q. crassifolia* Humb. & amp. Bonpl., and other broadleaved trees (*Servicios Técnicos Forestales STF, 2011*).

### Tree inventory and laboratory analysis

In 2022, 12 circular sites of 400 m$^2$ with slope compensations were established in a targeted manner in stands harvested during 2013–2014 under the methodology used by *Chávez-Pascual et al. (2017)* and *Miguel-Martínez et al. (2016)*. The sites were established with four silvicultural treatments: free thinning (FT), first thinning (T1), regeneration cut (RC) under the parent tree method and unmanaged area (UA), with three repetitions of each ST. The sites were geopositioned with a global positioning system (GPS) (Garmin eTrex 30, USA®) MAPS 6 within each stand. Descriptive variables of the site were taken: altitude (m); average slope of the site (%), with a clinometer (Haglöf®, EC II D-HS115, Spain); and exposure.

At the sites, a tree inventory was carried out, recording the variables: diameter at breast heigt (DAH, cm) with a diametric tape (Hartmann®, 283D); total height (TH, m) with a clinometer (Haglöf®, EC II D-HS115); and crown area (CA, m) with a tape measure. Trees

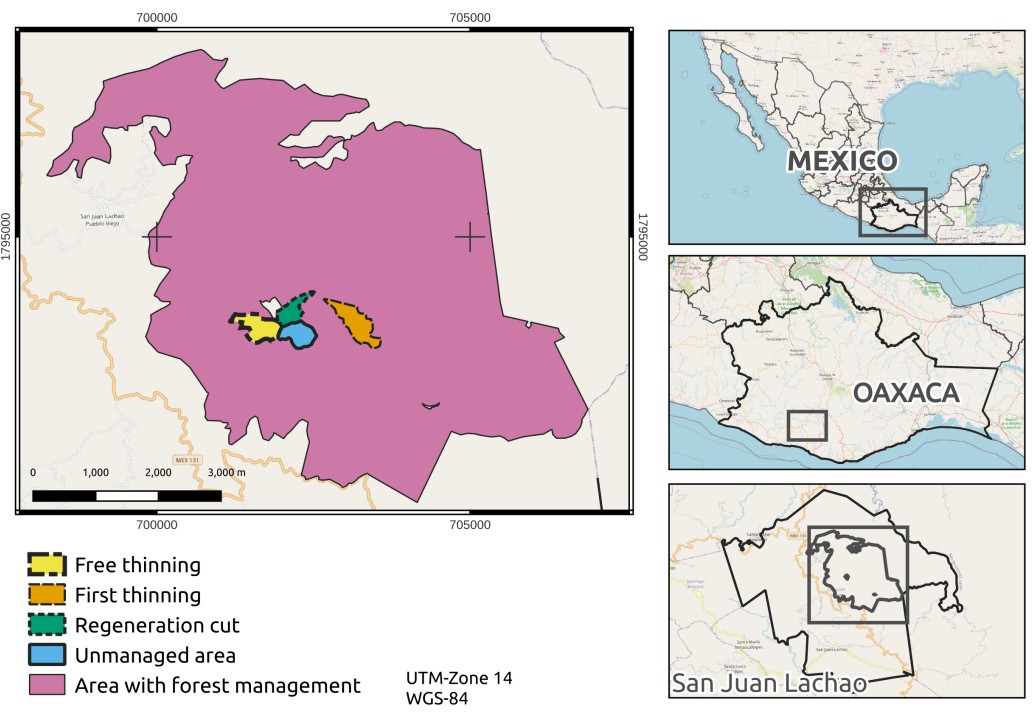

**Figure 1** Forest management area with sampling sites. Map credit: *OpenStreetMap Contributors, 2023*. Licensed under CC BY-SA 2.0.

were selected in proportion to the relative frequency of each species to obtain a sample (chip or slice); for *Pinus spp.*, a Pressler drill (Haglöf®, Sweden) was used to obtain a chip at a height of 1.30 m; in some species such as *Quercus spp.*, *Arbutus xalapensis* Kunth, and *Alnus acuminata* Kunth, at least one tree of each species was cut per site to obtain a five cm thick slice (*Rodríguez-Ortiz et al., 2019*).

The samples were analyzed in the Agroecosystems Laboratory of the Technological Institute of the Valley of Oaxaca. Core increment and slices of all species were weighed on an analytical scale (Shimadzu®, ATY224, ±1 mg) to determine their green weight (GW g). The green volume (GV, cm³) was determined with a digital Vernier caliper and by applying Newton's formula (*Romahn de la Vega & Ramírez, 2010*). Subsequently, the samples were placed into a drying oven (Memmert®, 100-800) at 102 °C until constant weight and dry weight (DW, g) were obtained. The specific gravity (SG, kg m⁻³) by species was obtained: SG = (DW/GV) × 100 (*Bhardwaj et al., 2016*). Subsequently, the samples were ground to determine their C content through the content of organic matter under the Walkey and Black method (*SEMARNAT, 2000*); 58% of the determined organic matter was C (Fig. 2).

The total tree volume with bark (TTVbark, m³) of all tree species found was determined with the forest biometric system (*Vargas-Larreta et al., 2017*); the product volume and SG (kgm⁻³) provided the TB content (kg tree⁻¹) (*Rodríguez-Ortíz et al., 2011*).

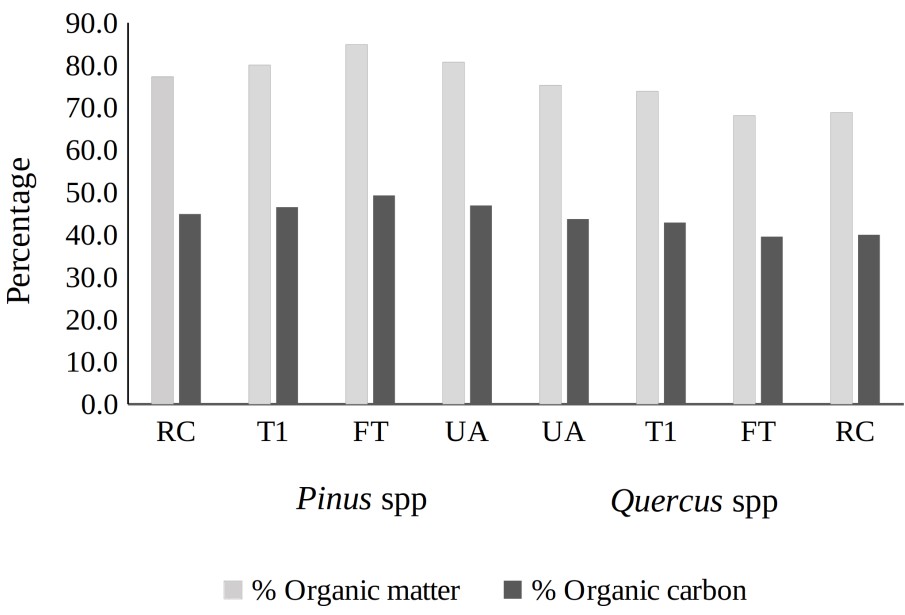

**Figure 2** **Organic matter and carbon content (%) of tree species in sites under silvicultural treatments.**
RC = regeneration cut, T1 = first thinning, FT = free thinning, UA = unmanaged area.

**Table 1** **Spectral band characteristics.**

| Number of band | Band name | Wavelenght (nm) | Resolution (m) |
|---|---|---|---|
| 1 | Coastal aerosol | 443.9 | 60 |
| 2 | Blue | 496.6 | 10 |
| 3 | Green | 560 | 10 |
| 4 | Red | 664.5 | 10 |
| 5 | Vegetation Red Edge | 703.9 | 20 |
| 6 | Vegetation Red Edge | 740.2 | 20 |
| 7 | Vegetation Red Edge | 782.5 | 20 |
| 8 | NIR | 835.1 | 10 |
| 8A | Narrow NIR | 864.8 | 20 |
| 9 | Water vapour | 945 | 60 |
| 10 | SWIR –Cirrus | 1373.5 | 60 |
| 11 | SWIR | 1613.7 | 20 |
| 12 | SWIR | 2202.4 | 20 |

## Sentinel 2 spectral data

The spectral image of the study area was downloaded, divided into 13 bands, on the Sentinels Scientific Data Hub platform (https://scihub.copernicus.eu/). The discharge was at level 2A, which involves preprocessing at the atmospheric correction level, (Table 1).

The image was processed with QGIS 2.16.16® software, where atmospheric correction was first applied to each band. To visualize the image in true color, a set of bands 2, 3 and 4 was constructed, and all three had a 10 m resolution per pixel. Subsequently, a set of bands
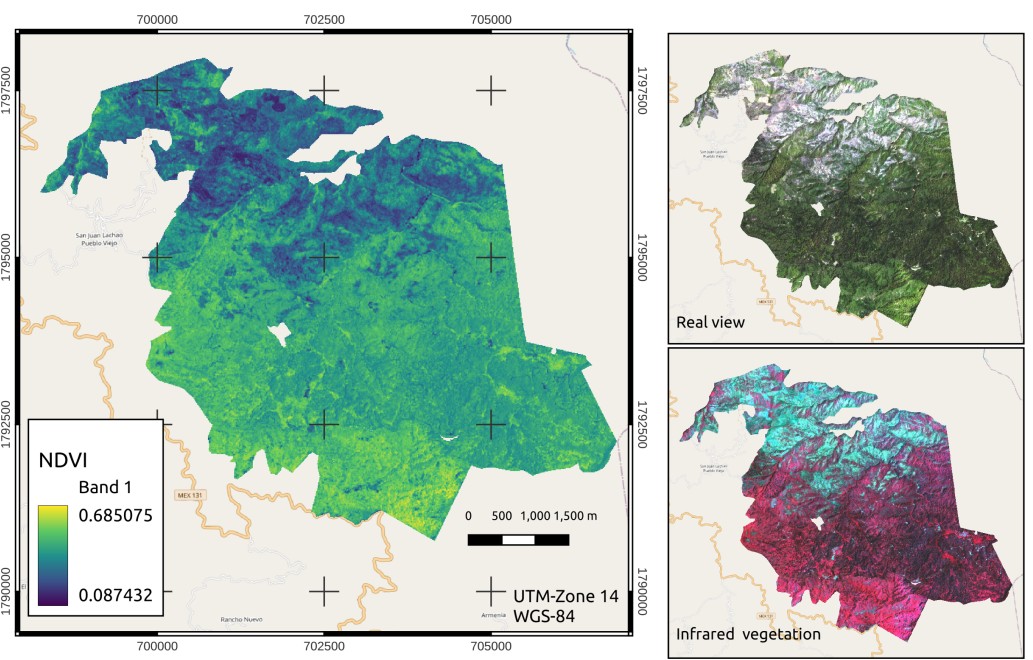

**Figure 3** **NDVI expansion, true color and near-infrared vegetation visualization of the managed area.** Map credit: Images were processed from Sentinel-2 data, *OpenStreetMap Contributors, 2023*, Licensed under CC BY-SA 2.0.

3, 4 and 8 (10 m pixel$^{-1}$) was constructed to visualize the vegetation in infrared bands (Fig. 3).

Bands 4 (red) and 8 (near infrared (NIR)) were used to calculate the NDVI per pixel. The following formula was applied: NDVI = (NIR –red)/(NIR + red), where red and NIR are the spectral reflectance measurements acquired in the red (visible) and near infrared regions, respectively. These spectral reflectances are proportions of the radiation reflected on each individual spectral band. The NDVI, real visualization and infrared vegetation layers were cut from the area under use and from the 4 stands where the sampling sites were located (Fig. 4).

Other vegetation indices were also tested in this study:

*GNDVI* = (NIR − Green)/(NIR + Green), where: GNDVI = Green Normalized Difference Vegetation Index, NIR =near infrared band, Green =green band.

*EVI* = 2.5 × ((NIR − Red)/((NIR) + (C1 × Red) − (C2 × Blue) + L)), where: EVI = Enhanced Vegetation Index, NIR =near infrared band, Red = red band, Blue = blue band, C1 and C2 = Atmospheric correction coefficients (6 and 7.5), L = soil influence correction factor.

*SAVI* = ((NIR − Red)/(NIR + Red + L)) × (1 + L), where: SAVI = Soil Adjusted Vegetation Index, NIR = near infrared band, Red = red band L = soil luminosity correction factor (0.428).

*AVI* = [NIR × (1 − Red) × (NIR − Red)]$^{1/3}$ , where: AVI =Advanced Vegetation Index, NIR =near infrared band, Red =red band.
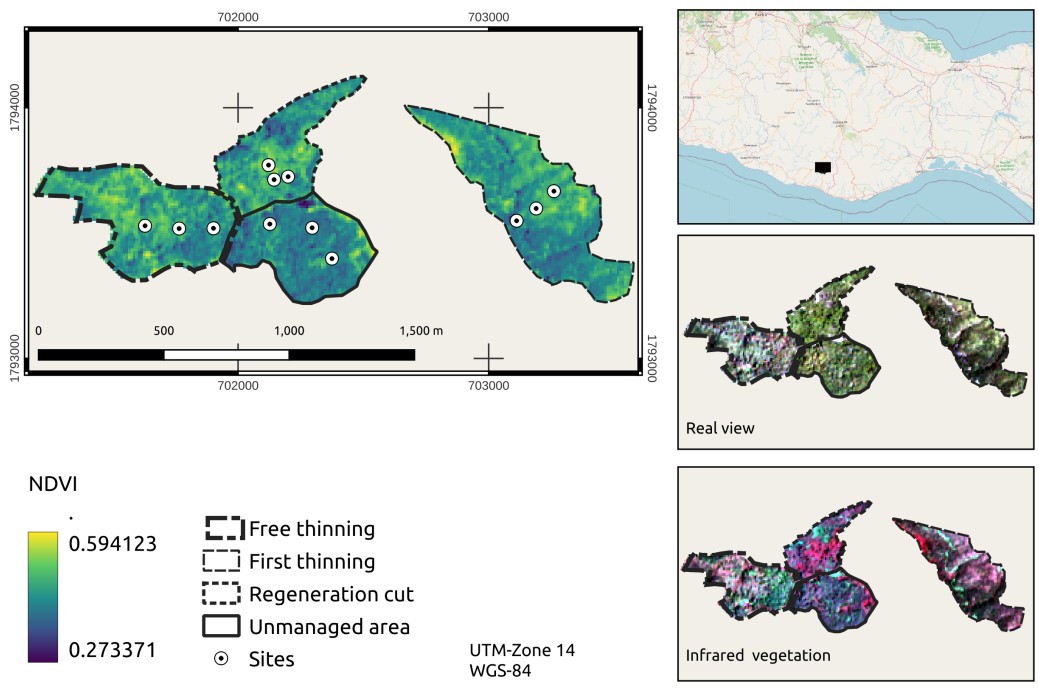

**Figure 4** NDVI expansion, true color and near-infrared vegetation visualization of managed stands.
Map credit: Images were processed from Sentinel-2 data, *OpenStreetMap Contributors, 2023*, Licensed under CC BY-SA 2.0.

## Data analysis

For the TB and C variables, the assumptions of normality and homogeneity of variances were verified (Shapiro–Wilk and Bartlett test, respectively, $\alpha = 0.05$). Once the NDVI values per pixel (10 m resolution) were obtained, the average value of the NDVI per sampling site was determined using a $3 \times 3$ pixel window surrounding the central point. For this part of the process, a validation was carried out through "raster statistics for points". Allometric models were adjusted to estimate C and TB (t pixel$^{-1}$ and t ha$^{-1}$) based on the NDVI (Table 2). Subsequently, with the models, the increase in TB and carbon was determined by site, stand and management area through the NDVI value of each pixel.

The STs were differentiated using a generalized linear model (PROC GLM) and the Tukey means test ($\alpha = 0.05$). All analyses were performed in the Statistical Analysis System (SAS) program (*SAS Institute, Inc., 2017*).

## RESULTS

### Model adjustment to estimate biomass and carbon

The linear models without an intercept ($\beta_0$) were those that best adjusted for the estimation of TB and C as a function of the NDVI, being highly significant ($p \leq 0.01$). The NDVI explained 92.2% and 91.9% of the existing variation in TB and C, respectively. The coefficients of variation (CVs) $\leq 32.2\%$ were moderate-high due to the variation in tree density and coverage between the applied STs. The standard deviation of the models was

**Table 2** Model fitting with different vegetation indices for biomass (B) and carbon (C) estimation. (t pixel$^{-1}$).

| Model | Parameter ($\beta_1$) | $R^2$ | $\sqrt{MSE}$ | CV (%) |
|---|---|---|---|---|
| $B = \beta_1 \times NDVI$ | 3.05115971** | 0.92 | 0.42** | 30.2 |
| $C = \beta_1 \times NDVI$ | 1.36719158** | 0.92 | 0.19** | 30.9 |
| $B = \beta_1 \times GNDVI$ | 3.3303334** | 0.92 | 0.42** | 30.2 |
| $C = \beta_1 \times GNDVI$ | 1.4923619** | 0.92 | 0.19** | 30.9 |
| $B = \beta_1 \times EVI$ | 3.0480304** | 0.91 | 0.44** | 31.7 |
| $C = \beta_1 \times EVI$ | 1.3648815** | 0.91 | 0.20** | 32.5 |
| $B = \beta_1 \times SAVI$ | 4.0386605** | 0.92 | 0.44** | 31.4 |
| $C = \beta_1 \times SAVI$ | 1.8087569** | 0.91 | 0.20** | 32.2 |
| $B = \beta_1 \times AVI$ | 3.4448735** | 0.92 | 0.42** | 30.2 |
| $C = \beta_1 \times AVI$ | 1.5436013** | 0.92 | 0.19** | 30.9 |

**Notes.**

Pixel, 100 m$^2$; $R^2$, coefficient of fit; MSE, Mean Square of Error; CV, coefficient of variation; NDVI, Normalized Difference Vegetation Index; GNDVI, Green Normalized Difference Vegetation Index; EVI, Enhanced Vegetation Index; SAVI, Soil Adjusted Vegetation Index; AVI, advanced vegetation Index.
**Highly significant ($t \leq 0.001$).

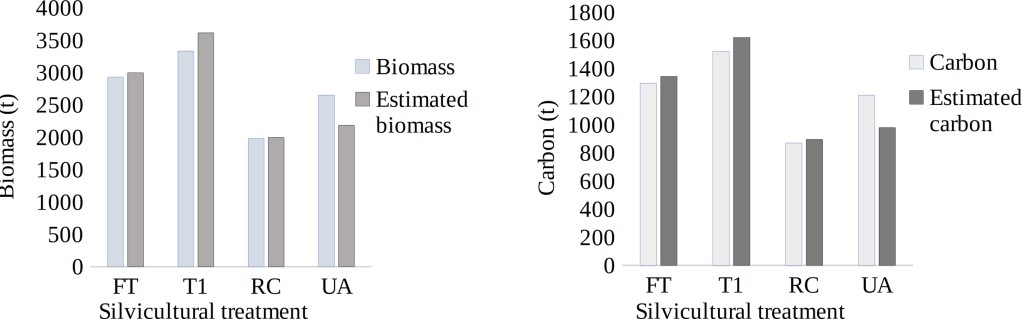

**Figure 5** Biomass (A) and carbon (B) per stand, estimated with field data *vs.* spectral data. RC = re-generation cut, T1 = first thinning, FT = free thinning, UA = unmanaged area.

low, which indicates that the estimated TB and C values were slightly removed from the trend of the mean (Table 2).

The model was validated through a Student's $t$ test ($\alpha = 0.05$) comparing TB and C per stand (t), estimated with field data *vs.* estimated with spectral data. No differences were found between both types of data ($p > 0.05$) (Figs. 5A and 5B). Therefore, the models were efficient at predicting TB and C through the NDVI since they generated reliable results when estimating at a large scale (estimation).

## Biomass and carbon estimations

For both variables, TB and CC, the stands under FT (21.4 ha) and T1 (26.4 ha) presented similar average estimated values of 140 t TB ha$^{-1}$ and 63 t C ha$^{-1}$ and 137 t TB ha$^{-1}$ and 61 t C ha$^{-1}$, respectively, such that the total amounts per the FT and T1 stands were 1,343 t C and 1,620 t C, respectively. The minimum and maximum values between the areas under

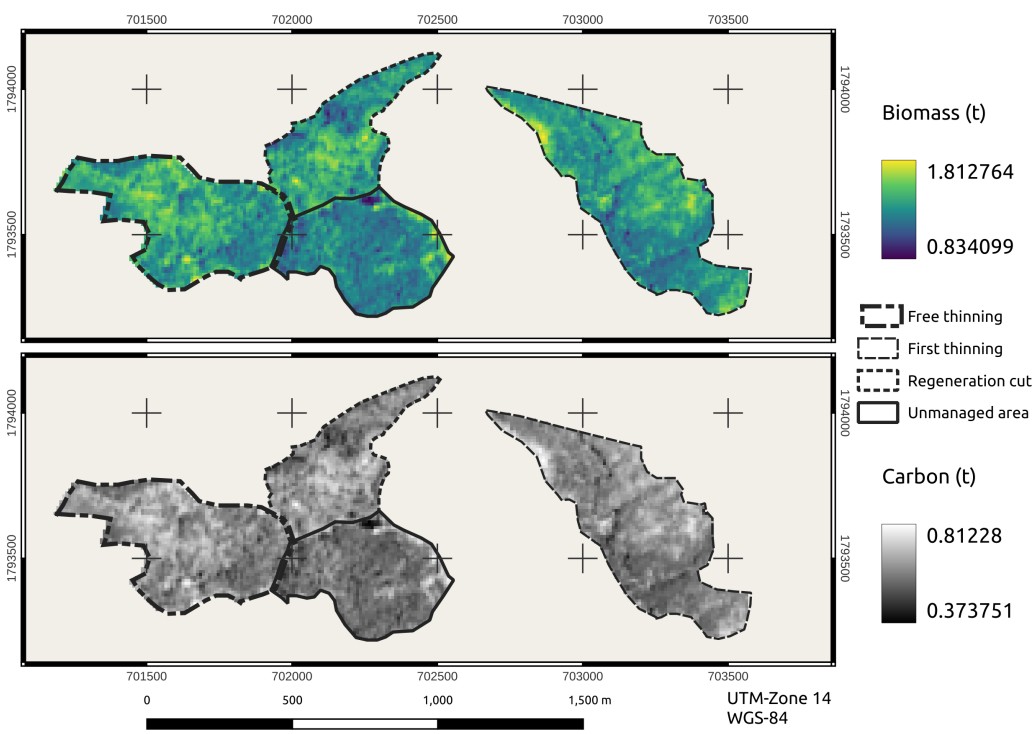

**Figure 6** **Estimations of tree biomass and carbon in stands under treatments.** Map credit: Images were processed from Sentinel-2 data.

the two types of STs were similar because their function was to decrease tree density when they are in the pole stage (Fig. 6).

The stand with RC (14.4 ha) presented values of 139 t TB ha$^{-1}$ and 62 t C ha $^{-1}$, with minimum and maximum values for C between 41 and 81 t ha$^{-1}$, showing a good accumulation of C in response to the regeneration of the parent tree treatment. The UA (17 ha) presented average values of 129 t TB ha$^{-1}$ and 58 t C ha$^{-1}$, with a range of variation between the minimum and maximum values of 37 to 79 t C ha$^{-1}$. The stands subjected to the STs (FT, T1, and RC) presented the same maximum value of carbon accumulation (81 t ha$^{-1}$), which was 2% higher than that in the UA (Fig. 6).

The FT stand presented 8.6% more carbon (t ha$^{-1}$) than the UA stand, the FT and UA stands had tree densities of 1166 trees ha$^{-1}$ and 825 trees ha$^{-1}$, respectively. This small difference in carbon values occurred because the FT stands maintained a contemporaneous structure with similar ND, crown diameter (CD) and TH, while the UA stands were highly heterogeneous, presenting variation in their area.

In the area managed by San Juan Lachao (3201 ha), a total of 404,048.08 t TB and 181,049.56 t C were estimated. An average accumulation of 126 t TB h$^{-1}$ and 57 t C ha$^{-1}$ was estimated, with minimum and maximum values of 27 to 209 t TB ha$^{-1}$ and 12 to 94 t C ha$^{-1}$, respectively. The areas with high accumulations of TB and C were observed in an intense red color, unlike the green areas, which showed the least accumulation (Fig. 7).

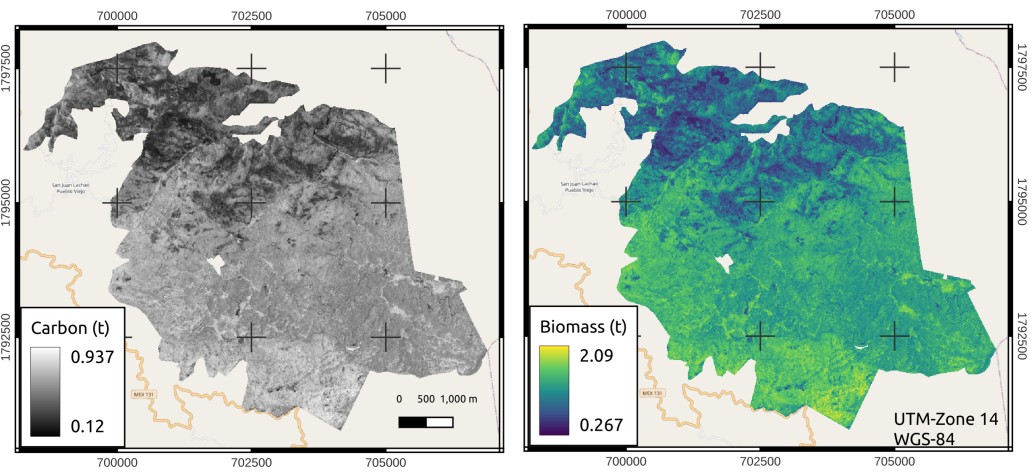

**Figure 7** Estimations of aboveground biomass and carbon in the managed area of San Juan Lachao.
Map credit: Images were processed from Sentinel-2 data, *OpenStreetMap Contributors, 2023*, Licensed under CC BY-SA 2.0.

**Table 3** Comparison of biomass and carbon estimated in the field *vs.* estimated with Sentinel 2.

| Variable | Silvicultural treatments | | | |
|---|---|---|---|---|
| (t ha$^{-1}$) | Free thinning | First thinning | RC | UA |
| B_*Pinus* spp | 65.9 ± 13.9[a] | 98.0 ± 7.4[a] | 112.8 ± 29[a] | 98.6 ± 32.0[a] |
| B_*Quercus* spp | 70.9 ± 19.0[a] | 28.2 ± 10.6[a] | 24.8 ± 5.8[a] | 57.7 ± 22.5[a] |
| B_tree | 136.8 ± 25.4[a] | 126.2 ± 16.5[a] | 137.5 ± 28.6[a] | 156.3 ± 28.4[a] |
| B_tree *Sentinel 2* [*] | 138.4 ± 4.2[ab] | 135.8 ± 3.2[ab] | 148.7 ± 3.6[a] | 129.2 ± 5.7[b] |
| C_*Pinus* spp | 32.4 ± 6.8[a] | 45.5 ± 3.4[a] | 50.6 ± 13.0[a] | 46.2 ± 15.0[a] |
| C_*Quercus* spp | 28.0 ± 7.5[a] | 12.1 ± 4.5[a] | 9.9 ± 2.3[a] | 25.2 ± 9.8[a] |
| C_tree | 60.4 ± 11.0[a] | 57.6 ± 7.3[a] | 60.4 ± 12.8[a] | 71.4 ± 13.2[a] |
| C_total *Sentinel 2* [*] | 62.0 ± 1.9[ab] | 60.9 ± 1.4[ab] | 66.6 ± 1.6[a] | 57.9 ± 2.5[b] |

**Notes.**
RC, regeneration cut (seed trees); UA, unmanaged area (conservation).
*Values from Normalized Difference Vegetation Index.
Different letters in rows indicate significant differences (Tukey, 0.05). Mean ± standard error.

All STs resulted in similar amounts ($p > 0.05$) of TB and C by genus (Pinus and Quercus), as well as total TB amounts (inferred in field sampling). On the other hand, the tree C estimated with the NDVI in the spectral image was higher ($p \leq 0.01$) in the RC (66.6 t ha$^{-1}$) than in the UA (57.9 t ha$^{-1}$), which showed greater heterogeneity; the same behavior was found for TB (Table 3).

This difference was visible because the UA had a high heterogeneity, so making an inference with field data was not feasible, unlike the estimation with the satellite image, which was more accurate in terms of irregular stand structure (Table 3). In addition, the areas with STs had a more homogeneous trend with a systematically distributed canopy, and the opposite occurred for the ASM, where heterogeneity led to an irregular structure.

## DISCUSSION

### Model adjustment

The stands with STs (FT, T1, and RC) showed similar TB and C values but varied from those of the UA due to the age of the stand, the crown area, the size of the trees, edaphic factors and altitude (*Rajput, Bhardwaj & Pala, 2017*; *Ruiz-Díaz et al., 2014*). The areas with T1 and FT thinnings presented an average of 138.5 t TB ha$^{-1}$ and 62 t ha$^{-1}$ of carbon. The area with RC presented 139 t TB ha$^{-1}$ and 62 t C ha$^{-1}$ (Fig. 6). Regarding the above information, *Aguirre-Salado et al. (2009)* reported in contemporary stands of *P. patula* Schl. *et* Cham. under harvested conditions, an estimate determined by remote sensing was 55 t C ha$^{-1}$.

The best fitted models in this work were "simple linear" as a function of the NDVI (10 m pixel$^{-1}$), showing significance ($p \leq 0.01$) and adjusted $R^2 = 0.92$, with which the total managed area (3201 ha) having 126 t TB ha$^{-1}$ (values from 27 to 209 t TB ha$^{-1}$) and 57 t C ha$^{-1}$ (values from 12 to 94 t C ha$^{-1}$), which shows potential values of TB and C (Fig. 7). *Thurner et al. (2014)* adjusted simple linear models to estimate carbon ($R^2 = 0.70$–$0.90$) in mixed forests of North America, Europe and Asia, where they found an average of $58 \pm 22.1$ t C ha$^{-1}$. *Reyes-Cárdenas et al. (2019)* estimated the forest TB in northern Mexico, adjusting an exponential model based on the NDVI, and found lower values (0.85 to 157 t ha$^{-1}$) than those in this study.

Choosing the best model depends on the type of vegetation that is being evaluated and the type of optical satellite that is being used; thus, the evaluator is responsible for choosing the best model; for example, *Aguirre-Salado et al. (2009)* adjusted a multiple regression model using the NDVI and the water stress index as independent variables ($R^2 = 0.70$). D'Oliveira et al. (2020) estimated the biomass of 10 forest plots in the southwestern Brazilian Amazon, adjusting multiple linear models based on LiDAR variables, ($R^2 = 0.90$, RMSE $= 13.23$ t ha$^{-1}$), presenting values from 11.1 to 273 t ha$^{-1}$. On the other hand, *Rex et al. (2020)* estimated biomass with LiDAR images (229.10 t ha$^{-1}$) in tropical forests with low intensity logging in Pará, Brazil, although in this case using ordinary least squares (OLS) regression and a selection of variables through principal component analysis ($R^2 = 0.35$ - $R^2 = 0.53$), and the model had less significance than that of other.

The model was validated through Student's $t$ test ($\alpha = 0.05$) comparing TB and C per stand (t), and the values were estimated with field data *vs.* estimated with spectral data, showing no significant differences ($p > 0.05$); correlating field values with spectral data provides effective and accurate information (*Vaghela et al., 2021*). *Reyes-Cárdenas et al. (2017)* correlated field data with spectral data and obtained similar values. *Aguirre-Salado et al. (2012)* adjusted a linear model to estimate aerial biomass based on the NDVI and continuous vegetation fields (CVFs) (adjusted $R^2 = 0.77$ CME $= 26.00$ t ha$^{-1}$), showing high validation correlation coefficients ($r = 0.87^{**}$); *Verly et al. (2023)* estimated the C of the Atlantic Forest with Sentinel 2 images, showing a high correlation coefficient between the estimated and observed carbon averages ($r = 0.84$).

## Biomass and carbon estimation

The UA presented TB and C values of 129 and 58 t ha$^{-1}$ (Fig. 6), respectively, which indicated that it is necessary to use satellite images for estimations since they provide values that can be extrapolated to the area of study. *Perea-Ardila, Andrade-Castañeda & Segura-Madrigal (2021)* found similar TB and tree C values as a function of the NDVI in a high Andean forest of Boyacá, Colombia, determining $168.0 \pm 11.2$ t TB ha$^{-1}$ and $84.0 \pm 5.61$ t C ha$^{-1}$. Similarly, *Clerici et al. (2016)* reported $180.7 \pm 23.8$ t ha$^{-1}$ of biomass in 400 m$^2$ plots in Andean forests in Cundinamarca, and *Yepes-Quintero et al. (2011)* reported TB and CC of $102.38 \pm 25.22$ t ha$^{-1}$ and $51.19 \pm 12.61$ t ha$^{-1}$, respectively, in highland forests of Antioquia. On the other hand, *Bhardwaj et al. (2016)* reported an estimated biomass with the NDVI of 169.05 to 265.83 t ha$^{-1}$ in subtropical forests of the northwestern Himalayas. Therefore, satellite images are a potential tool for differentiating managed and unmanaged areas (*Avogadro & Padró, 2019*).

After testing various vegetation indices in this work, the NDVI was best adjusted (Table 2) to estimate TB and CC (0.09–0.69), providing positive values and indicating areas with healthy and vigorous vegetation (Fig. 3); however, although in some areas the NDVI values were low, the values of TB and CC were positive. This result was in contrast to those of *Aguirre-Salado et al. (2012)*, where the estimates tended to be negative when the pixels had low NDVI values. The NDVI is an excellent indicator because it had three visible red bands that increased accuracy and had the near infrared band (*Wang et al., 2018*). Similarly, their the estimates of C for areas under timber management were highly accurate given the methodology that was selected was correctly followed (*Yan et al., 2016*).

The NDVI to estimate TB and C in this work was crucial, since the NIR spectrum contains information about the chemical composition (infrared light energy) through the photosynthetic pigment "chlorophyll"; chemical bond information such as C-H, N-H, S-H, $C = O$ and O-H (*Guerrero-Maestre et al., 2008*). For this reason, the NDVI was adapted to the type of vegetation evaluated, where it has a high reflectance in the near infrared and low in the red band (*Jensen, 2015*).

Although there are other methods to evaluate the vegetation, such as the use of LiDAR (Light Detection and Ranging) images, in this study the most feasible was the use of satellite images, due to the scale to be evaluated; and it is suitably viable since it provides us with information on areas with and without timber forest management. Using Sentinel 2 satellite images in vegetated areas significantly increases estimate accuracy (*Polat, Akcay & Balik Sanli, 2022*).

Through remote sensing, it is possible to extract information about the vegetation in pine-oak forests that are unmanaged and classify the vegetation more easily that through observation methods (*Ancira-Sánchez & Treviño Garza, 2015*; *Polat, Akcay & Balik Sanli, 2022*). However, factors such as topography, altitude, slope, precipitation, and temperature must always be considered, as indicated in *Olthoff, Martinez-Ruiz & Alday (2016)*. By determining the TB and CC of a large UA, this area can be classified into smaller areas, and suitable silvicultural management methods and treatments can be determined through a management plan (*Agus et al., 2004*; *Puletti, Chianucci & Castaldi, 2018*), in addition to promoting C cycle mechanisms (*Herold et al., 2011*).

## CONCLUSIONS

In comparison to estimating TB and CC with field data, estimating them with spectral data in a forest area that is unmanaged resulted in a greater degree of precision because the estimates were based on information from the NDVI of the entire surface of the stand, while those based on field data, where only inferences are made, were less precise. Therefore, using satellite images to classify new areas of forest use is feasible, and these images can be used for monitoring before and after a cutting cycle and collaborating on carbon credit mechanisms, which benefit society and support future generations.

## ACKNOWLEDGEMENTS

We thank the community authorities and forestry technical services of the community of San Juan Lachao, Juquila, Oaxaca, Mexico.

### Funding
The authors received no funding for this work.

### Competing Interests
The authors declare there are no competing interests.

### Author Contributions
- Ashmir Ambrosio-Lazo conceived and designed the experiments, performed the experiments, analyzed the data, prepared figures and/or tables, authored or reviewed drafts of the article, and approved the final draft.
- Gerardo Rodríguez-Ortiz conceived and designed the experiments, performed the experiments, analyzed the data, prepared figures and/or tables, authored or reviewed drafts of the article, and approved the final draft.
- Joaquín Alberto Rincón-Ramírez conceived and designed the experiments, performed the experiments, analyzed the data, prepared figures and/or tables, authored or reviewed drafts of the article, and approved the final draft.
- Vicente Arturo Velasco-Velasco analyzed the data, prepared figures and/or tables, authored or reviewed drafts of the article, and approved the final draft.
- José Raymundo Enríquez-del Valle analyzed the data, prepared figures and/or tables, authored or reviewed drafts of the article, and approved the final draft.
- Judith Ruiz-Luna analyzed the data, prepared figures and/or tables, authored or reviewed drafts of the article, and approved the final draft.

### Data Availability
The data is available in the Supplemental File.

## Supplemental Information

Supplemental information for this article can be found online at http://dx.doi.org/10.7717/peerj.16431#supplemental-information.

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
