# Peer review of "Carbon mapping in pine-oak stands under timber management in southern Mexico"

_PeerJ, doi:10.7717/peerj.16431_

## Round 0.1 · original submission · Minor Revisions

The referees both found merit in the work. They have raised in their reports some issues that require attention but this should be possible by undertaking relatively minor revision. As noted in the reviews aspects of the writing could be enhanced and please ensure correct use of subscripts and superscripts.

·

Basic reporting

Introduction:
General comment: language and phrasing need revision.

Specific comment:
Satellite sensors measure vegetation indices>>> not very clear, need a phrasing (line 55)

This paper introduces only the importance of NDVI, what about the other VIs such as GNDVI, EVI, SAVI, and AVI used in this research? It’s a good way to start with the definition of the VIs, their implication in the Aboveground biomass and carbon stock measurement, modeling, and predicting, and the reason for selecting certain VIs used in this study (lines 70 to 75).

Materials and methods:
General comment: language and phrasing need revision.
Specific comments:
Need revision on the way of writing the scientific name of the species (lines 101 and 103)

Results
Fairly ok

Discussion
Discussion on the potentiality of the red edge band of sentinel-2 in biomass estimation, the saturation problem in the NDVI, and its effect on biomass and carbon estimation is crucial.

The discussion section should also give the reasons behind the differences found and it should try to explain if there are other, potentially better solutions to the problem. Another thing is discussing the limitations of the study as well as the potential for other studies to solve such limitations.

The calculation and prediction of AGB are examples of regression problems that machine learning has a lot of potentials to address. I am aware that the authors have chosen a class of models to test, but at the very least, the potential enhancements provided by machine learning algorithms like SVM, RF, and ANN DT should be discussed and cited to provide a wider perspective on potential methods.

Conclusion
Fairly ok

Experimental design

Materials and methods:

Mention the VIs formula for the GNDVI, EVI, SAVI, and AVI as they were used for the analysis too (lines 158 - 161)

The model fitting formula of R2, RMSE, and C.V (%) needs to be added if possible as it gives the general overview of the model fitting to the readers

Shapiro-Wilk and Bartlett test, respectively, alpha = 0.05, for the normality and homogeneity p-value must be greater than 0.05, please have a look at this (line 166)

The sample plot of 400m2 cannot be compared with the pixel size of the sentinel (10m x 10m). please JUSTIFY

What is the procedure to obtain per pixel values needs to explain (lines 166 and 167)?

Validity of the findings

no comment

Additional comments

1. Spelling should be Symbology instead of simbology in all figure section

2. Scale bar units on the map are different. Uniformity must be maintained.

·

Basic reporting

1. The Study highlights a study conducted in southern Mexico that aimed to evaluate tree biomass (TB) and carbon content (CC) in underutilized forest stands using satellite images derived from Sentinel 2. Their main argument is that the traditional destructive and empirical methods for estimating carbon pools in managed timber forests are time-consuming, expensive, and impractical on a large scale. Therefore, the researchers turned to satellite imagery as a cost-effective and efficient alternative for assessing carbon stocks.

2. In 2022, the researchers established 12 circular sites with four different silvicultural treatments (STs): 1st thinning (T1), free thinning (FT), regeneration cut (RC), and an unmanaged area (UA). A tree inventory was conducted, and samples were collected to determine TB using specific gravity and CC using the Walkley and Black method. The researchers obtained a satellite image of the study area from Sentinel 2 and used it to develop a simple linear model based on the Normalized Difference Vegetation Index (NDVI) at a resolution of 10 meters per pixel. The model showed significance and a high R2 adj value of 0.92, indicating a strong relationship between NDVI and TB/CC.

3. Using the developed model, the researchers estimated the TB and CC (in metric tons per hectare) for each ST and managed area. The total managed area (3,201 hectares) had an average TB of 126 t/ha and CC of 57 t/ha. Among the different ST areas, the free thinning (FT) treatment exhibited the highest TB accumulation (140 t/ha) and CC (63 t/ha) without significant differences compared to the unmanaged area (UA), which had 129 t/ha TB and 58 t/ha CC. These findings suggest that proper implementation of silvicultural treatments maintains a balance in tree carbon accumulation. The study concludes that satellite images from Sentinel 2 provide reliable estimates of TB and CC in managed forest stands. The study demonstrates the potential of satellite imagery for assessing carbon stocks in large-scale forest management, offering a cost-effective and efficient alternative to traditional methods.
The study meet the standards of the jouranal and there is no improvement required.

Experimental design

Based on the information provided, the study's methodology appears to be coherent and understandable, effectively conveying the main points and findings. There are no significant design problems evident in the provided information.

Validity of the findings

Based on the study's design, sample representativeness, data collection methods, and statistical analyses undertaken in this study, I can say that the findings are valid.

Additional comments

The study is novel, and its novelty lies in its combination of utilizing satellite imagery for estimating carbon pools, focusing on underutilized stands, comparing silvicultural treatments, and addressing the cost and time efficiency challenges associated with traditional methods. These aspects contribute to the advancement of knowledge and understanding in the field of forest carbon assessment and management.

---

## Round 0.2 · Minor Revisions

Thank you for your revised manuscript and the response to the reviewers' comments. The revisions appear to have enhanced the article but the quality of the text is a concern. While the meaning is typically clear the article would benefit from further revision to enhance the quality of the text. The Section editor has requested that professional proofreading be undertaken and proof of this having been undertaken submitted with the revised manuscript. Such revision will greatly enhance the impact of your work.

**Language Note:** The Academic Editor has identified that the English language must be improved. PeerJ can provide language editing services - please contact us at copyediting@peerj.com for pricing (be sure to provide your manuscript number and title). Alternatively, you should make your own arrangements to improve the language quality and provide details in your response letter. – PeerJ Staff

---

## Round 0.3 · accepted · Accept

Thank you for revising the article and providing details on the language editing. I shall recommend your article for acceptance.